# The Cost-Effectiveness Analysis of Gamiguibi-Tang versus Hwangryunhaedok-Tang for Patients with Insomnia Disorder Based on a Randomized Controlled Trial

**DOI:** 10.3390/healthcare10112157

**Published:** 2022-10-28

**Authors:** Ji-Eun Lee, In-Chul Jung, So-Young Lee, Jung-Hwa Lim, Bo-Kyung Kim, Eun Cho

**Affiliations:** 1College of Pharmacy, Sookmyung Women’s University, Seoul 04310, Korea; 2Department of Oriental Neuropsychiatry, College of Korean Medicine, Daejeon University, Daejeon 34520, Korea; 3Department of Neuropsychiatry, School of Korean Medicine, Pusan National University Korean Medicine Hospital, Pusan National University, Yangsan 50612, Korea

**Keywords:** cost-effectiveness analysis, Gamiguibi-tang, herbal medicine, insomnia disorder, productivity loss

## Abstract

(1) Insomnia is associated with poor quality of life and loss of productivity, and is a significant economic burden on society. Gamiguibi-tang (GGBT) is the most frequently prescribed herbal medicine for insomnia treatment. Hwangryunhaedok-tang (HHT) is used as an insured herbal medicine for insomnia in the Korean National Health Insurance (NHI) system. This study aims to evaluate the cost-effectiveness of GGBT versus HHT in patients with insomnia disorders based on clinical trial data; (2) Methods: The EuroQol five-dimension scale (EQ-5D) was used to estimate quality-adjusted life-years (QALY). Direct and non-direct medical costs and lost productivity costs were estimated. The cost-effectiveness of GGBT was compared with HHT treatments over six weeks from a societal perspective; (3) Results: A total of 81 patients who underwent GGBT (*n* = 56) and HHT (*n* = 25) treatment completed the clinical trial. The EQ-5D score improved significantly more in the GGBT than in the HHT group (0.02 vs. −0.03, *p* < 0.05). The QALYs for six weeks were slightly greater in GGBT (0.0997) than in the HHT group (0.0987); however, the total costs incurred were approximately 9% less in GGBT ($934) than in the HHT group ($1029). GGBT was found to be a more economically dominant treatment option compared to HHT for treating insomnia; (4) Conclusions: Among herbal medicines, GGBT may be a cost-effective option for treating insomnia from a societal perspective in Korea.

## 1. Introduction

Insomnia is a common public health problem among the elderly, women, workers, and people with mental illness [1,2]. Insomnia disorder is diagnosed in case of sleep dissatisfaction, such as difficulty initiating or maintaining sleep or waking up early in the morning, despite adequate opportunities and circumstances to sleep [3,4]. Approximately 30% of the general population has more than one symptom of insomnia [5,6]. Due to poor sleep quality, patients with insomnia have numerous impairments in the daytime, such as fatigue, mood disturbance, and loss of concentration/memory [7]. These negative consequences of insomnia are associated with decreased quality of life (QoL) [8,9,10], productivity loss [8,9,10,11], and an increased risk of occupational and motor vehicle accidents [11]. Furthermore, insomnia is a risk factor for the worsening of various psychiatric and physical comorbidities, including depression [12,13], anxiety disorder [12], dementia [14], cardiovascular disease [15], and the development of suicidal intentions [16].

The societal cost burden of insomnia is significant, with combined direct and indirect costs in the United States of America (USA) estimated at US $100 billion per year [17]. Productivity loss is an important reason for the economic burden of insomnia [18,19,20,21,22], as the annual cost of lost productivity was 70% greater for insomnia patients ($1739) than for good sleepers ($1013) in 2005 [21]. The lost productivity costs were related to patients who missed work (absenteeism) or decreased on-the-job performance (presenteeism) due to insomnia [8]. The costs associated with presenteeism are particularly significant [17,18,20]. For example, the annual cost of presenteeism was about five times higher than that of absenteeism and approximately 24 times higher than the treatment costs [18]. The conditions coexisting with insomnia are another reason for the socioeconomic burden because insomnia patients with medical comorbidities increase healthcare utilization and generate direct and indirect costs [17,23,24]. Therefore, therapeutic interventions that improve sleep quality and productivity and manage comorbidities can reduce the financial burden of insomnia disorder in society [17,25].

Cognitive-behavioral therapy for insomnia (CBT-i) and short-term hypnotics are recommended forms of treatment for insomnia [26,27,28]. However, in the real clinical setting, CBT-I has been underutilized owing to a lack of clinician training and its time-intensive nature [29]. More frequently utilized are long-term hypnotics with severe side effects, including tolerance, addiction, and dependency [26,29]. Given the limitations of standard therapies, complementary and alternative medicine (CAM), without serious side effects, could be another therapeutic option for insomnia [30,31,32,33]. For a long period, herbs such as valerian and chamomile have traditionally been considered sleep inducers and sedatives [31,34]. Even today, natural products containing herbal ingredients are most commonly used as CAM therapies for sleep complaints [33]. Patients having sleep disturbance can also choose several forms of CAM, including exercise (e.g., meditation, yoga, Tai Chi), body-based practices (e.g., massage, acupressure), and dietary supplements (e.g., vitamin probiotics) [30,31,32]. More than 1.6 million individuals in the USA use CAM therapies to improve insomnia or sleep problems [33]. Korean medicine (KM) is a representative CAM option for the treatment of insomnia [35]. Approximately 12% of Korean patients with insomnia are treated with KM alone or in parallel with Western medicines [35,36]. KM is particularly used by patients who are dissatisfied with sleeping pills or want to reduce the dose [37]. In KM, acupuncture and/or herbal medicines are commonly used to treat insomnia patients [37].

Several Korean herbal formulas have been used to treat insomnia, and such regimens have been prepared and applied according to the patient’s pattern identification [37]. Gamiguibi-tang (GGBT; Kamikihito in Japanese; Jiaweiguipi-tang in Chinese) is used for insomnia patients with heart and spleen deficiencies due to mental stress and excessive tension [38,39]. Guibi-tang (Kihito in Japanese; Guipi-tang in Chinese), the basis of GGBT, has been the most frequently prescribed traditional herbal formula for treating insomnia in East Asia for hundreds of years [37,39,40]. In Korea, more than half of KM doctors choose the Guibi decoction first for severe and chronic insomnia, according to a survey examining the practice patterns of KM doctors [37]. GGBT is composed by adding *Gardenia Fruit* and *Bupleurum Radix* to Guibi-tang (*Ginseng Radix, Atractylodis Rhizoma Alba, Longanae Arillus, Astragali Radix, Angelicae Gigantis Radix, Polygalae Radix, Glycyrrhizae Radix et Rhizoma, Zizyphi Fructus, Zingiberis Rhizoma, Hoelen, Aucklandiae Radix, Zizyphi Semen, Moutan Cortex Radicis*), and is available in the form of extract granule as well as a decoction. GGBT granules have been approved by the Korea Food and Drug Safety Administration (KFDA) for insomnia, anemia, anxiety, and nervousness in people with poor complexion due to weakness. Although GGBT granules are not currently covered by the Korean National Health Insurance (NHI), they are widely used for insomnia.

In 1987, nationwide insurance coverage was applied to KM to ensure patients’ access to KM services, including herbal preparations, acupuncture, moxibustion, and cupping. The current NHI reimbursement scheme includes 68 single-herbal preparations and 56 herbal formulas [41]. However, the herbal formulas covered by NHI, accounting for only 14% of the total formulas published in KM prescription textbooks, are insufficient to cover various conditions [41,42]. Compared to other countries (337 in Taiwan [43], 147 in Japan [44], and 987 formulas in China [45]), the coverage level is relatively limited in Korea. Accordingly, uninsured herbal medicines extracted from various herbal formulas have been frequently used, and their coverage under NHI has been demanded by KM clinicians [41,46].

Although GGBT is regarded as an empirically standardized herbal formula for insomnia in traditional oriental medicine [39,40,47], scientific evidence to support whether GGBT treatment is reasonable remains scarce. Only one clinical trial has demonstrated that GGBT administration for two weeks can significantly decrease insomnia severity; however, this study targeted patients with cancer-related sleep disturbance [47]. In addition, the economic advantages of GGBT have not yet been evaluated in Korea. To obtain insurance coverage for GGBT granules in the future, it may be necessary to compare the cost-effectiveness of GGBT with other herbal medicines currently under coverage for insomnia. Among 56 insurance formulas in Korea, Hwangryunhaedok-tang (HHT, or Oren-gedoku-to in Japanese, or Huanglian-jie-du in Chinese) is a herbal preparation proven effective in insomnia in Korea [48]. HHT, a popular detoxication formula, can be used for insomnia related to fire/heat patterns by dissipating excess heat and fire toxins [48]. HHT consists of four crude herbs: *Scutellaria baicalensis, Gardenia fruit, Coptis chinensis Rhizom,* and *Phellodendrom Bark*.

This study was performed to determine the cost-effectiveness of GGBT compared to HHT in patients with insomnia. This study aimed to provide economic evidence for using GGBT in treating insomnia disorders compared to HHT. In particular, considering the significant economic burden caused by productivity loss due to insomnia, this study examined and compared the productivity loss costs between GGBT and HTT treatments.

## 2. Materials and Methods

### 2.1. Clinical Trial Design and Participants

This clinical trial was a multicenter, randomized, double-blind, active comparator-controlled trial involving patients with insomnia. To obtain more preliminary data on the safety and effectiveness of GGBT, patients were randomly assigned to the GGBT or HHT group in a 2:1 ratio. A total of 96 participants (64 in GGBT; 32 in HHT) were required, based on an effect size of 0.69, a significance level of 0.05, a power of 0.80, and a dropout rate of 20%. GGBT (3 g tid) or HHT (1.87 g tid) were administered orally to the treatment and control groups three times daily for three weeks from the start date. The packaging of both herbal medicines was identical, opaque, and thick, making it difficult for the participants to see the contents. Patients visited the institution weekly during the three-week treatment period to perform outcome assessments and check for vital signs and adverse events. Three weeks after the end of treatment, the participants visited once again to evaluate the follow-up observations. The primary outcome was measured by Insomnia Severity Index (ISI) and the change in score from baseline to 3 weeks. The secondary outcome was assessed by Pittsburgh Sleep Quality Index (PSQI), sleep diary, actigraphy, Visual Analog Scale (VAS), Korean Symptom Checklist-95 (K-SCL-95), and QoL.

According to protocol, the inclusion criteria for participants were as follows: an ISI of at least 15, age between 19 to 80 years, and diagnosis of insomnia disorders according to the Diagnostic and Statistical Manual of Mental Disorders Fifth edition (DSM-5) criteria. Concurrent use of hypnotics was allowed, but the relevant drug and dosage needed to be maintained for at least four weeks before the start date of the clinical trial. For the exclusion criteria, individuals treated with any KM or non-pharmaceutical treatments (e.g., CBT-i, meditation) for insomnia during the four weeks before the start date were not allowed to enroll in the clinical trial. In addition, participants with other medical comorbidities (e.g., major depression, panic, alcohol disorder, and hypertension) were excluded. Shift workers were also excluded from the clinical trial.

### 2.2. Effectiveness Measurement

Patients with insomnia were asked to complete questionnaires on QoL and clinical efficacy measures at baseline, at the end of treatment (3 weeks), and at the end of the follow-up period (6 weeks). The EuroQol five-dimension scale (EQ-5D) was used to estimate QoL. EQ-5D health states were converted to utility weight using the South Korean tariff [49], and the values were used to calculate the quality-adjusted life years (QALYs) for six weeks by adopting the area under the curve (AUC).

The clinical effectiveness outcomes were assessed in terms of the ISI and PSQI. ISI is a useful instrument for assessing insomnia treatment [50] and is a reliable and valid index in Korea (Cronbach’s alpha = 0.92) [51]. The total score ranges from 0 to 28, with higher scores associated with greater insomnia severity. The PSQI measures the quality of sleep during the previous month [52] and is a standardized and validated index in Korea (Cronbach’s alpha = 0.84) [53]. The PSQI score ranges from 0 to 21, with higher scores indicating poorer sleep quality.

Work productivity and activity impairment (WPAI) was used to quantify work productivity loss and regular daily activity impairment. It is a validated instrument to measure the amount of health-related productivity loss during the past seven days [54] and is a sensitive measure for insomnia [55]. This instrument consists of six items that assess absenteeism (work time missed due to health problems), presenteeism (degree of decrease in on-the-job performance due to health problems), overall work productivity loss (combination of absenteeism and presenteeism), and activity impairment (degree of decrease in daily activities due to health problems) score [56]. Only paid employees can provide the work productivity loss score. In this study, overall work productivity loss and activity impairment scores were suggested. Both productivity loss score values were estimated to range from 0 to 1, with higher values indicating a more negative impact of insomnia on productivity.

### 2.3. Costs Measurement

The cost types considered in this economic evaluation model were direct medical and non-medical and lost productivity costs. Direct medical costs included the costs generated by the clinical trial protocol, including herbal medicine costs, preparation fees, first-visit or revisit diagnosis, and examination fees. Direct non-medical costs included round-trip transportation fares to visit KM hospitals during clinical trials. The unit of travel costs per outpatient visit to a KM hospital was estimated by converting previous data of the average transportation fees according to the type of medical institution in 2008 [57] to 2021 Korean won (KRW) monetary values [58].

The productivity loss costs were calculated by multiplying the productivity loss scores obtained through the WPAI questionnaire at baseline and three- and six weeks with hourly compensation (i.e., hourly wage or value of unpaid labor). The wage value was obtained from the monthly average wage and increase/decrease by labor type from the Korean Statistical Information Service (KOSIS) [59]. The employment rate was estimated based on data from all participating subjects and was assumed to be equal for the GGBT and HHT groups. The value of household work was estimated based on a report published in 2014 (KRW 10, 569 per hour) [60]. The hourly value in 2014 was converted to the 2021 KRW monetary value by adjusting the average growth rate of the value of unpaid household service work [61]. The average hours of household work were estimated based on the 2019 Time Use Survey data [62]. All costs were calculated in KRW for the reference year 2021. Discounting was not necessary because the study period was 6 weeks.

### 2.4. Cost-Effectiveness Analysis

The cost-effectiveness model was established from a societal perspective, where the QALYs of treatment were considered for effectiveness, and all costs associated with insomnia and treatment were considered. Two cost-effectiveness outcomes were also assessed. First, the average cost-effectiveness ratio (ACER), the average cost-effectiveness of each therapeutic intervention, was estimated based on a cost-effectiveness analysis. Second, the incremental cost-effectiveness ratio (ICER) for GGBT over HHT was estimated.

This study evaluates two additional models by varying the cost types considered. In Model A, the daily activity impairment cost, considered a productivity loss cost in the primary model, was excluded. In Model B, only medical costs (direct medical and non-medical costs) were considered, and the cost-effectiveness result was interpreted from the healthcare perspective.

### 2.5. Statistical Analysis

This economic evaluation was performed using per-protocol data from patients who completed 3 weeks of herbal treatment and had the primary endpoint result of the randomized control trial (RCT), which was the difference in ISI score change before and after the 3-week herbal treatment. Baseline characteristics were analyzed descriptively and are presented as means, standard deviations, or frequencies. An independent t- or chi-squared test was used to confirm the homogeneity of the baseline characteristics between the GGBT and HHT groups. A paired *t*-test was performed to assess improvement in treatment outcomes (i.e., EQ-5D, ISI, PSQI, WPAI) between baseline and endpoint within the groups. The outcome changes between the baseline and 6-week study endpoints were compared between the GGBT and HHT groups by using an independent *t*-test. The significance level was set at 0.05 (two-tailed). All statistical analyses were performed using the IBM SPSS Statistics version 25 (IBM Corp., Armonk, NY, USA).

## 3. Results

A total of 96 patients were randomly assigned to the GGBT (*n* = 64) and HHT (*n* = 32) groups in a 2:1 ratio. Among them, 81 subjects (*n* = 56 in GGBT, *n* = 25 in HHT) who completed all 6 weeks of the study period were included in the economic analysis. The CONSORT flowchart of the study process is shown in Figure 1. The baseline characteristics of the study participants are presented in Table 1. There were no statistical differences between the groups for any variable at baseline (Table 1). A total of 14 (25%) adverse events were reported in the GGBT group and 12 (48%) in the HHT group. Mild adverse events such as nausea, vomiting, or diarrhea were observed, but neither definitely related nor serious adverse events were observed.

### 3.1. Treatment Outcomes

In the GGBT group, the EQ-5D increased slightly over 6 weeks from 0.848 to 0.870, but it was not statistically significant (*p* = 0.092, Table 2). In the HHT group, EQ-5D continually decreased from 0.869 to 0.838 during the study period, with no significance (*p* = 0.165). The changes in EQ-5D from baseline to the 6-week follow-up point were statistically significantly different between the GGBT and HHT groups (*p* = 0.031) (Table 2). Based on the EQ-5D scores over 6 weeks, QALYs were estimated to be 0.0997 for GGBT and 0.0987 for HHT (Table 3).

At the end of treatment (3 weeks), the ISI and PSQI scores decreased compared with the baseline scores in both the GGBT and HHT groups (Table 2). At the follow-up point (6 weeks), the GGBT group showed a further decrease for both ISI and PSQI. However, in the HHT group, both ISI and PSQI scores bounced up at the follow-up observation (Table 2). The changes in ISI (*p* = 0.306) and PSQI (*p* = 0.699) scores during the six weeks were not significantly different between the GGBT and HHT groups (Table 2).

In the GGBT group, the overall work productivity loss score significantly decreased from 0.407 to 0.249 (*p* = 0.008) at 3-week and to 0.223 (*p* = 0.002) at 6-week point. In the HHT group, the score decreased from 0.37 to 0.33 (*p* = 0.785) at 3-week and to 0.30 (*p* = 0.414) at follow-up (Table 2). Activity impairment scores decreased at three weeks compared to baseline for both the GGBT (*p* = 0.000) and HHT groups (*p* = 0.004). However, the activity impairment scores were increased at follow-up in the GGBT groups while maintaining a lower level compared to the baseline score (Table 2). The changes in work productivity loss (*p* = 0.357) and activity impairment score (*p* = 0.694) between the baseline and 6 weeks showed no significant intergroup differences (Table 2).

### 3.2. Costs Outcomes

The costs incurred during the 6 weeks after initiating treatment are shown in Table 3. The direct medical costs for traditional Korean treatment were greater in GGBT (KRW 339,000) than in HHT (KRW 244,000) because the unit cost of GGBT (KRW 5600 per day) was 5.6 times greater than that of HHT (KRW 1000 per day). The direct non-medical costs were identical in both groups. The work productivity loss cost was slightly greater by 34% in the HHT (KRW 312,000) than in the GGBT (KRW 232,000). The impairment cost in regular daily activity was also 25% greater for HHT (KRW 610,000) than for GGBT (KRW 486,000). The total cost of six weeks in the HHT group (KRW 1,177,000) was slightly greater (10%) than that in the GGBT group (KRW 1,069,000) (Table 3).

The productivity loss cost during the six weeks before enrollment in the clinical trial was almost similar between the GGBT and HHT participants (Figure 1). After enrollment, the productivity loss cost was reduced by 35.2% and 18.3% in the GGBT and HHT groups, respectively. The total costs related to insomnia after receiving GGBT (KRW 1,069,000) were lower than the costs for six weeks before the intervention (KRW 1,109,000), while the total costs in the HHT group increased after treatment from KRW 1,129,000 to KRW 1,177,000 (Figure 2). The main reason for cost savings in the GGBT group was the reduction in lost productivity costs.

### 3.3. Cost-Effectiveness Outcomes

Table 4 summarizes the results of the cost-effectiveness analysis. The ACER values were KRW 10,722,000 and KRW 11,920,000 per QALYs for the GGBT and HHT, respectively (Table 4). Due to lower costs incurred in the GGBT group than those in the HHT group, GGBT treatment is considered an economically dominant and cost-saving option compared to HHT treatment for insomnia disorders.

For modified Model A, which excluded daily activity impairment costs, the incremental cost per QALY was 17,211,000 KRW/QALY (Table 4). In Model B, where only medical costs were considered among the cost types from the healthcare perspective, the ICER was 100,420,000 KRW/QALY (Table 4).

## 4. Discussion

In the present study, an economic evaluation was performed to compare the cost-effectiveness of GGBT granules and HHT powder in patients with insomnia disorders from a societal perspective regarding improvement in QoL, clinical effectiveness, and cost outcomes. This economic evaluation demonstrated that the GGBT was a cost-effective treatment for insomnia compared to HHT because the GGBT group occurred less cost of productivity loss than the HHT group. In EQ-5D, ISI, PSQI, and WPAI measurements, the GGBT group improved more than the HHT group for six weeks.

As patients with insomnia have worse QoL than good sleepers, it is important to improve their QoL through treatment interventions [8,17]. As a generic QoL instrument, EQ-5D, which measures QoL in this study, may lack sensitivity in detecting insomnia-specific QoL changes [63]. Similar to the EQ-5D results, ISI and PSQI, which are insomnia-specific clinical indicators, also improved in the GGBT group for 6 weeks. Considering this, GGBT can be interpreted as a potential herbal medicine to enhance the QoL of patients with insomnia. Unlike GGBT, the QoL of participants in the HHT group continued to decrease during the treatment and follow-up periods. The clinical effects of ISI and PSQI in the HHT group increased but decreased at follow-up. It may be suggested that the maintenance effect of GGBT on patients’ QoL was greater than that of HHT.

Productivity improvement induced by herbal medicines was confirmed using the WPAI in this study, which was widely used to measure productivity loss in Korea [57]. This instrument was also utilized to compare the productivity loss in people with and without insomnia in large national surveys conducted in the USA and Europe [8,9,10,64]. The participants’ overall work productivity and daily activity impairment scores at the baseline of this study (overall work productivity loss scores of 0.41 in GGBT, 0.37 in HHT, and activity impairment scores of 0.49 in GGBT, 0.53 in HHT) were similar to the WPAI scores of patients with insomnia in previous studies (overall work productivity loss scores of 0.25–0.40, and the daily activity impairment scores of 0.4–0.5) [8,9,10]. The result that GGBT reduced the work productivity loss score by three times and daily activity impairment score by 1.2 times more than the HHT during 6 weeks suggested that GGBT might be a more promising treatment than HHT for enhancing productivity in patients with insomnia.

This study presents the economic outcomes of herbal medicines from a societal and healthcare system perspective to help decision-makers in Korea evaluate the suitability and reasonability of GGBT benefits in the future. In the base model, which included medical costs and costs related to productivity loss in work and daily activity from a societal perspective, GGBT was considered an economically favorable treatment compared with HHT. Modifying the base model by considering only work productivity loss for the indirect cost category, as in most other economic evaluations [65,66], the ICER result was found to be within the acceptable threshold range of KRW 35,200,000 (as GDP per capita in 2021) per QALY. In the subsequent analysis of the model from a healthcare perspective, the ICER result exceeded the willingness to pay in South Korea. However, the results revealing that insomnia treatment alternatives are more cost-effective from a societal than from a healthcare perspective were similar to previous economic evaluations of insomnia [66,67]. This suggests that cost-saving of lost productivity is the main driver of economic gains through insomnia treatment. Hence, in terms of socioeconomic burden, it is necessary to increase the accessibility of GGBT treatment rather than that of HHT treatment.

The main strength of this study is the fact that the elements of lost productivity costs were exclusively estimated in an effort to avoid underestimating the economic effects of GGBT. In the early 2000s, most insomnia studies considered only absenteeism to evaluate productivity loss cost [19,22]. However, presenteeism has also been linked to the cost burden of insomnia [18]. In addition, insomnia has been significantly associated with daily task impairment [8,9,10]. Thus, recent insomnia studies have expanded the concept of productivity loss costs by considering presenteeism [18,20,21,65,66] and unpaid labor-related productivity loss [18,67]. In the present study, the majority of the participants (76%) were not economically active populations; they might have been unpaid workers or housekeepers. Hence, the results of this study, including costs associated with absenteeism, presenteeism, and unpaid productivity loss, exclusively comprise the full societal burden and cost-effectiveness consequences of treating insomnia patients with herbal medicines.

Another strength of the present study is that it is the first economic evaluation that supplemented the scientific evidence of GGBT, a CAM therapy widely used for insomnia in Korea. Despite its high prevalence and considerable cost burden, insomnia disorder is an under-treated condition [68,69]. Insomnia patients do not seek medical help because of a lack of awareness of treatment options and the perception of therapeutic interventions as ineffective [70]. CAM therapy has a few outcome studies to support its effectiveness, and these studies have often been evaluated as poor quality [27]. High-quality evidence is needed to understand and determine the value of CAM regimens in insomnia patients and among clinicians [32,39]. The present study provides evidence that GGBT has a positive effect on QoL and productivity, and it might help increase the population’s confidence in GGBT treatment for insomnia and the accessibility of KM.

Despite the aforementioned advantages of this study, several limitations must be considered while interpreting the results. First, the random assignment of patients in the RCT might not be identical and applicable to the general practice setting of KM. In reality, whether to take a GGBT or HHT regimen is determined by the pattern identification of insomnia. GGBT is appropriate for patients with deficiencies related to the heart and spleen [38,39], whereas HHT is suitable for individuals who need detoxification [37]. However, the randomization approach, which does not consider the cause of insomnia and the patient’s constitutional type, differs from the general practice of KM and may weaken the validity of the results. Further practical clinical and observational studies need to be conducted to reflect practical settings and identify the unbiased clinical efficacy of GGBT. However, considering that the HHT regimen has already been covered and utilized as a herbal medicine without restriction, the present results suggest that the GGBT regimen can also be covered by insurance for patients with general insomnia.

Second, the costs expended on Western medicine were missed in the present study. Indeed, the majority of participants in this study could not accurately remember the number of visits to a Western hospital and the names and dosages of the hypnotics consumed, thereby providing unreliable information. However, the medical costs related to Western medicine may not have differed between the two groups because the proportion of patients using Western medicine was similar at baseline unless the treatment regimen was changed during the treatment period.

Finally, the period of this study (6 weeks) was relatively short compared to previous economic evaluations of other insomnia treatment alternatives [65,66,67]. There was no statistically significant difference between the groups, despite the fact that the incremental benefit of GGBT treatment over 6 weeks was greater than that of HHT in terms of ISI, PSQI, WPAI, and QALYs. This is because the study period was not long enough to present a difference in treatment outcomes considering the features of insomnia as a chronic condition. However, as GGBT was considered to have a better maintenance effect than HHT, if the QALYs could be calculated beyond six weeks, the net QALYs between the groups may be widened.

The recent COVID-19 pandemic has raised the risk of depression and anxiety, which are highly associated with insomnia in almost all Organisation for Economic Co-operation and Development (OECD) countries compared to the pre-COVID times [71,72]. While the prevalence of depression and anxiety has been higher than the OECD average, and the suicide rate has been ranked the highest in Korea [71], the quality of treatment for mental illness is inferior to that available in other countries [73]. Therefore, an effective treatment for insomnia that can help relieve linked psychiatric disorders and reduce the cost burden is urgently needed. It has been reported that patients with insomnia who are concerned about the side effects of hypnotics tend to seek KM treatment rather than visit Western medical clinics [37]. Lack of knowledge about cost-effective herbal medicine for insomnia may hamper reasonable decision-making by individual clinicians, patients, or policymakers responsible for rational healthcare resource allocation. For example, the current circumstances in which HHT is an insured herbal medicine for insomnia without any supportive scientific evidence may limit the potential use of GGBT. As this study showed that GGBT granules are an economically favorable treatment option compared to HHT powder, it is suggested that GGBT might be a more suitable choice for patients with insomnia.

In conclusion, uninsured GGBT may be a more economically dominant herbal medicine than HHT, which is covered by the NHI in Korea. The cost savings from productivity loss is the main driver for reducing the economic burden related to insomnia. Furthermore, GGBT may be a successful intervention for improving the severity of insomnia disorder, sleep quality, and QoL.

## Figures and Tables

**Figure 1 healthcare-10-02157-f001:**
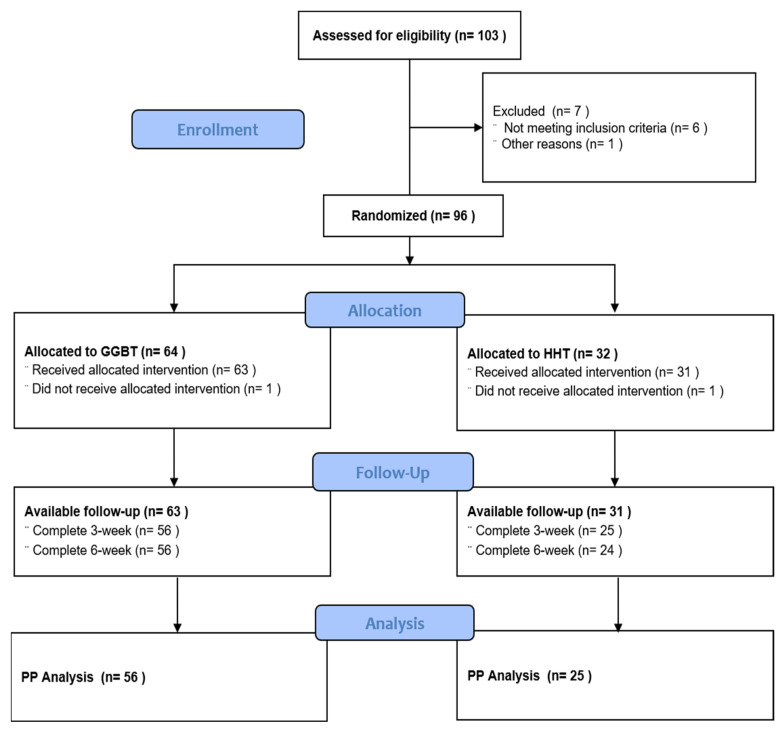
The CONSORT flowchart of study process.

**Figure 2 healthcare-10-02157-f002:**
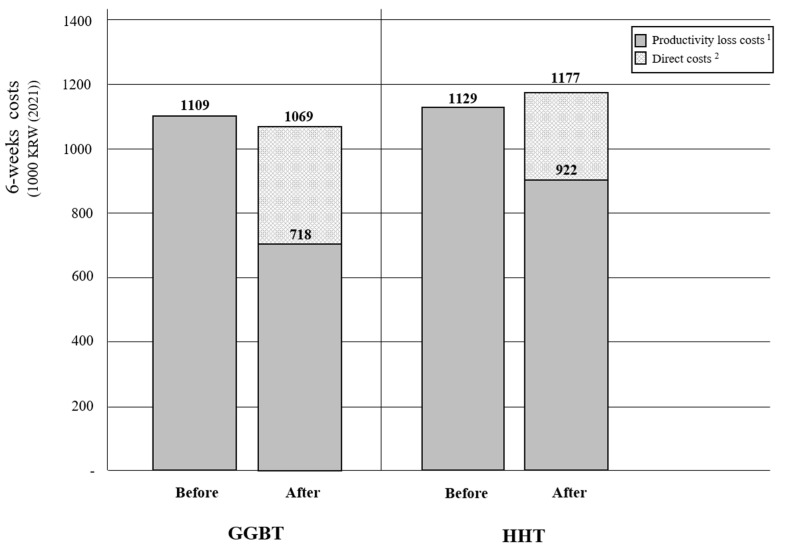
Comparison of 6 weeks per patient (1000 KRW (2021)) before and after initiating treatment. Abbreviations: GGBT, Gamiguibi-tang; HHT, Hwangryunhaedok-tang. ^1^ Productivity loss costs include costs for both work productivity loss and daily activity impairment. ^2^ Direct costs associated with traditional Korean treatment include direct medical and non-medical costs.

**Table 1 healthcare-10-02157-t001:** Baseline characteristics for participants.

	GGBT (*n* = 56)	HHT (*n* = 25)	*p*-Value ^1^
Age (mean ± SD)	61.96 ± 12.99	56.04 ± 15.93	0.081
19–29 [*n*(%)]	3 (5.4%)	3 (12%)	
30–39	3 (5.4%)	1 (4%)	
40–49	1 (1.8%)	3 (12%)	
50–59	6 (10.7%)	4 (16%)	
60–61	29 (51.8%)	10 (40%)	
70+	14 (25%)	4 (16%)	
Sex [*n*(%)]			
Male	22 (39.3%)	11 (44.0%)	0.690
Female	34 (60.7%)	14 (56.0%)	
Smoking [*n*(%)]			
Yes	5 (8.9%)	3 (12.0%)	0.669
No	51 (91.1%)	22 (88.0%)	
Drinking [*n*(%)]			
Yes	16 (28.6%)	6 (24.0%)	0.669
No	40 (71.4%)	19 (76.0%)	
Exercise [*n*(%)]			
Yes	47 (83.9%)	20 (80.0%)	0.666
No	9 (16.1%)	5 (20.0%)	
Prescribed hypnotics [*n*(%)]			
Yes	19 (33.9%)	5 (20%)	0.205
No	37 (66.1%)	20 (80%)	
EQ-5D score (mean ± SD)	0.848 ± 0.117	0.869 ± 0.104	0.449
ISI score (mean ± SD)	18.20 ± 4.08	18.28 ± 4.84	0.936
PSQI score (mean ± SD)	12.77 ± 2.98	11.96 ± 3.22	0.275
WPAI Work productivity loss (mean ± SD) ^2^	0.407 ± 0.21	0.367 ± 0.06	0.804
WPAI Activity impairment (mean ± SD) ^3^	0.489 ± 0.22	0.530 ± 0.19	0.407

Abbreviations: EQ-5D, EuroQol five-dimension scale; GGBT, Gamiguibi-tang; HHT, Hwangryunhaedok-tang; ISI, Insomnia Severity Index; PSQI, Pittsburgh Sleep Quality Index; SD, standard deviation; WPAI, work productivity, and activity impairment questionnaire. ^1^
*p*-value was calculated using the independent t- or chi-squared test. In the case of WPAI overall work productivity loss, the Mann-Whitney U test was used. ^2^ Among employed respondents (GGBT = 15, HHT = 3 at baseline). ^3^ Among respondents who completed the WPAI questionnaire (GGBT = 56, HHT = 23 at baseline).

**Table 2 healthcare-10-02157-t002:** Treatment outcomes within the group during follow-up and between groups (per protocol).

		GGBT (*n* = 56)			HHT (*n* = 25)			*p*-Value ^2^
		Mean	SD	*p*-Value ^1^	Mean	SD	*p*-Value ^1^	
EQ-5D (range −0.171–1)	Baseline	0.848	0.117		0.869	0.104		
	3 weeks	0.869	0.136		0.858	0.106		
	Change (95% CI) ^3^	0.021 (−0.00, 0.05)	0.096	0.104	−0.010 (−0.03, 0.02)	0.068	0.465	0.100
	6 weeks	0.870	0.128		0.838	0.166		
	Change (95% CI) ^3^	0.022 (−0.00, 0.05)	0.098	0.092	−0.031(−0.08, 0.01)	0.108	0.165	0.031
ISI (range 0–28)	Baseline	18.20	4.08		18.28	4.84		
	3 weeks	12.09	5.19		12.76	4.54		
	Change (95% CI) ^3^	6.11 (4.75, 7.50)	5.20	0.000	5.52 (3.58, 7.47)	4.71	0.000	0.631
	6 weeks	11.82	4.82		13.33	4.67		
	Change (95% CI) ^3^	6.38 (5.06, 7.69)	4.93	0.000	5.08 (2.71, 7.46)	5.62	0.000	0.306
PSQI (range 0–21)	Baseline	12.77	2.99		11.96	3.22		
	3 weeks	11.09	3.57		10.24	2.98		
	Change (95% CI) ^3^	1.68 (0.85, 2.51)	3.10	0.000	1.72 (0.71, 2.73)	2.44	0.002	0.953
	6 weeks	10.75	3.62		10.42	3.22		
	Change (95% CI) ^3^	2.02 (1.17, 2.87)	3.18	0.000	1.71 (0.24, 3.18)	3.48	0.025	0.699
WPAI Overall work productivity loss ^4^ (range 0–1)	Baseline	0.407	0.21		0.367	0.06		
	3 weeks	0.249	0.16		0.333	0.32		
	Change (95% CI) ^3^	0.158 (0.05, 0.26)	0.19	0.008	0.033 (−0.73, 0.79)	0.31	0.785	0.675
	6 weeks	0.223	0.18		0.300	0.17		
	Change (95% CI) ^3^	0.185 (0.10, 0.27)	0.16	0.002	0.067 (−0.31, 0.45)	0.15	0.414	0.357
WPAI Activity impairment ^5^ (range 0–1)	Baseline	0.489	0.22		0.530	0.19		
	3 weeks	0.318	0.19		0.426	0.22		
	Change (95% CI) ^3^	0.171 (0.12, 0.23)	0.21	0.000	0.104 (0.04, 0.17)	0.16	0.004	0.169
	6 weeks	0.354	0.24		0.417	0.19		
	Change (95% CI) ^3^	0.136 (0.07, 0.20)	0.24	0.000	0.113 (0.02, 0.20)	0.20	0.016	0.694

Abbreviations: EQ-5D, EuroQol five-dimension scale; GGBT, Gamiguibi-tang; HHT, Hwangryunhaedok-tang; ISI, Insomnia Severity Index; PSQI, Pittsburgh Sleep Quality Index; SD, standard deviation; WPAI, Work Productivity and Activity Impairment Questionnaire. ^1^
*p*-value was based on the paired *t*-test at baseline and post-treatment (at 3 or 6 weeks). For the WPAI overall work productivity loss score, *p*-value was obtained using the Wilcoxon signed-rank test. ^2^
*p*-value was computed using an independent *t*-test for changes between baseline and at 3 weeks or 6 weeks. In the case of WPAI overall work productivity loss, the Mann-Whitney U test was used to compare between-group changes. ^3^ Change was the mean difference between baseline and at 3 weeks or 6 weeks. ^4^ Among respondents who were employed (GGBT = 15, HHT = 3). ^5^ Among respondents who completed the survey (GGBT = 56, HHT = 23).

**Table 3 healthcare-10-02157-t003:** The results of quality-adjusted life years and costs for 6 weeks.

	GGBT	HHT	Difference
Utility (mean (SD))			
QALY (reference 0.1154) ^1^	0.0997 (0.01)	0.0987 (0.13)	0.0010
Costs (1000 KRW (2021))			
(1) Direct medical cost	339	244	96
(2) Direct non-medical cost	11	11	-
(3) Work productivity loss cost	232	312	−79
(4) Daily activity impairment cost	486	610	−124
Total costs (base model) ^2^	1069	1177	−108
Total costs (model A) ^3^	583	567	16
Total costs (model B) ^4^	351	255	96

Abbreviations: GGBT, Gamiguibi-tang; HHT, Hwangryunhaedok-tang. ^1^ The QALYs for 6 weeks of perfect health is 0.1154. ^2^ Base model included direct medical and non-medical costs and productivity loss costs at work and daily activities from a societal perspective. ^3^ Model A included direct medical and non-medical costs and cost of work productivity loss, except costs related to daily activity impairment from a societal perspective. ^4^ Model B included only direct medical and non-medical costs from the healthcare perspective.

**Table 4 healthcare-10-02157-t004:** The average and incremental cost-effectiveness ratios for GGBT compared to HHT.

	ACER, 1000 KRW per QALYs (USD per QALY) ^1^		ICER, 1000 KRW per QALYs (USD per QALYs) ^1^
Perspective	GGBT	HHT	
Base model ^2^	10,722 ($9372)	11,920 ($10,420)	Dominant
Model A ^3^	5848 ($5112)	5739 ($5016)	17,211 ($15,045)
Model B ^4^	3516 ($3074)	2582 ($2257)	100,420 ($87,780)

Abbreviations: ACER, average cost-effectiveness ratio; GGBT, Gamiguibi-tang; HHT, Hwangryunhaedok-tang; ICER, incremental cost-effectiveness ratio; QALYs, quality-adjusted life years. ^1^ 1$USD = 1144 KRW (2021). ^2^ The base model included direct medical and non-medical costs and productivity loss costs at work and daily activity from a societal perspective. ^3^ Model A included direct medical and non-medical costs and cost of work productivity loss, except costs related to daily activity impairment from a societal perspective. ^4^ Model B included only direct medical and non-medical costs from a healthcare perspective.

## Data Availability

The data analyzed during this study are not publicly available, as they were collected and analyzed in a clinical trial.

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
