# Peer review of "The Cost-Effectiveness Analysis of Gamiguibi-Tang versus Hwangryunhaedok-Tang for Patients with Insomnia Disorder Based on a Randomized Controlled Trial"

_healthcare, 2022, doi:10.3390/healthcare10112157_

Round 1
Reviewer 1 Report
With respect to the manuscript “The Cost-Effectiveness Analysis of Gamiguibi-Tang versus Hwangryunhaedok-Tang for Patients with Insomnia Disorder Based on a Randomized Controlled Trial”, I think that:
- It is very interesting.
- The subject is relevant for health services.
- The authors use adequate methodology.
However, I think that various aspects need to be improved:
- Authors do not have any reference to the approval of any ethical committee. It would be essential to be published.
- The inclusion/exclusion criteria are confused and should be clarify.
-It would be useful to have a figure with the experimental design of the study
- The doses of the herbal medicines administered to the patients are not mentioned in the material and methods.
-Although it was reported that adverse effects were monitored, it was not reported if adverse effects occurred during treatment with herbal medicines
- It is not referred in the manuscript if the questionnaires used in the study are validated for Korean population.
- Taking into account the limitations of the study and the fact that the only statistically significant parameter refers to quality of life, authors should be more cautelous in the conclusions.
Author Response
Thanks to the reviewer's comment.
As suggested, we modified and clarified our manuscript.
Please see the attachment.

Reviewer 2 Report
Comments to Authors
Comments to: The cost-effectiveness analysis of Gamiguibi-Tang versus Hwangryunhaedok-Tang for patients with insomnia disorder based on a randomized controlled trial
The author conducted a cost-effectiveness analysis to evaluate the cost-effectiveness of Gamiguibi-Tang (GGBT) versus Hwangryunhaedok-Tang (HHT) in patients with insomnia disorders from a randomized controlled trial. They found that the EQ-5D score improved more in GGBT than in the HHT group and the estimate quality-adjusted life-years (QALY) was also slightly greater in GGBT group. However, the total costs incurred were 9% less in GGBT group. They concluded that GGBT may be a cost-effective option for treating insomnia. This was an interesting study and the results could provide an important reason to choose of GGBT in clinic, but I also have some concerns regarding this study.
1. The author use one paragraph (paragraph 3 in the Introduction section) to introduce the cognitive-behavioral therapy for insomnia (CBT-i) that I think it was not necessary. For the main purpose of this study was to explore the effectiveness of herb medicine, so the Introduction should mainly focus on this.
2. In the Materials and Methods, the age range was between 19-80 years (this was a quite a large range), however the mean age was about 61 years and 56 years in the two groups. I want to know the age distribution of the included participants.
3. According to the CONSORT guideline, a patient flow chat was needed.
4. According to the CONSORT guideline, the randomization and blinding were also needed in the Material and Methods section.
5. The reason why the author choose the ration of 2:1 for GGBT and HHT group was not clear. And the reason to choose HHT as a controlled group was also not clear. Whether GGBT and HHT has different taste?
6. The sample size estimation and also the primary and secondary outcomes of this study was also needed.
7. I suggested to calculate the mean difference and 95%CI between baseline and 3 weeks or 6 weeks in Table 2.
8. This first paragraph of the Discussion should be the main results of this study.

Author Response
Thanks to the reviewer's comment.
As suggested, we revised and clarified the manuscript.
Please see the attachment.

Round 2
Reviewer 2 Report
Comments to Authors
Comments to: The cost-effectiveness analysis of Gamiguibi-Tang versus Hwangryunhaedok-Tang for patients with insomnia disorder based on a randomized controlled trial
The authors have addressed all my concerns and I have no other comments.